# Therapeutic Choice in Older Patients with Acute Myeloid Leukemia: A Matter of Fitness

**DOI:** 10.3390/cancers12010120

**Published:** 2020-01-02

**Authors:** Raffaele Palmieri, Giovangiacinto Paterno, Eleonora De Bellis, Lisa Mercante, Elisa Buzzatti, Fabiana Esposito, Maria Ilaria Del Principe, Luca Maurillo, Francesco Buccisano, Adriano Venditti

**Affiliations:** 1Hematology, Department of Biomedicine and Prevention, University Tor Vergata, 00133 Rome, Italy; raffaele.f.palmieri@gmail.com (R.P.); g.paterno@aol.com (G.P.); debellis.eleonora.1@gmail.com (E.D.B.); lisa.mercante@gmail.com (L.M.); buzzattielisa@gmail.com (E.B.); fabiana.e91@gmail.com (F.E.); dlpmlr00@uniroma2.it (M.I.D.P.); francesco.buccisano@uniroma2.it (F.B.); 2Fondazione Policlinico Tor Vergata, 00133 Rome, Italy; luca.maurillo@uniroma2.it

**Keywords:** acute myeloid leukemia, fitness, therapeutic choices

## Abstract

Acute myeloid leukemia (AML), with an incidence increasing with age, is the most common acute leukemia in adults. Concurrent comorbidities, mild to severe organ dysfunctions, and low performance status (PS) are frequently found in older patients at the onset, conditioning treatment choice and crucially influencing the outcome. Although anthracyclines plus cytarabine-based chemotherapy, also called “7 + 3” regimen, remains the standard of care in young adults, its use in patients older than 65 years should be reserved to selected cases because of higher incidence of toxicity. These adverse features of AML in the elderly underline the importance of a careful patient assessment at diagnosis as a critical tool in the decision-making process of treatment choice. In this review, we will describe selected recently approved drugs as well as examine prognostic algorithms that may be helpful to assign treatment in elderly patients properly.

## 1. Introduction

Acute myeloid leukemia (AML), with a yearly incidence in Europe of 5–8 cases per 100,000 individuals, is cancer predominantly of the elderly, with a median age at diagnosis of 67 years [1]. In the last decades, unlike for young adults, overall survival (OS) has not changed meaningfully for the elderly ones, with less than 10% of patients older than 65 years being alive 5 years after the diagnosis [2]. Considering that the life expectancy of people aged 65 in the Western world is of approximately 15–20 additional years, the negative social impact of AML is evident [3]. The reasons for poor outcomes of older patients can be attributed to both patient- and disease-related features. Advanced age is frequently associated with low performance status (PS), comorbidities, and organ impairment, most of them likely to evolve into organ failures, discouraging the use of intensive remission-oriented therapies [4]. Moreover, it is far more frequent for elderly patients to develop treatment complications, such as severe infections and prolonged hematological toxicities [5]. In addition to that, the biological and cytogenetic profile of elderly patients with AML differs from that of the young with a major incidence of unfavorable cytogenetics [6], of secondary AML supervening after previous hematological disorders like myelodysplastic syndrome (MDS) and myeloproliferative neoplasm (MPN), or following exposure to radio-chemotherapy (therapy-related AML) [5]. All these elements explain the poor outcome of AML of older/elderly patients, pointing out the importance of delivering as personalized as possible treatment strategies supplanting the conventional “one-size-fits-all” approach. Accordingly, there is an urgent need to identify appropriate tools that help physicians determine the candidacy of older patients for intensive or non-intensive approaches. This is an incredibly important decision, especially in the light of the assumption that “being fit” has always been regarded as a surrogate for “chance for survival”, since “fitness” is the green light to intensive chemotherapy delivery. Therefore, we will discuss the “pros and cons” of the algorithms that are in use to explore the geriatric vulnerability of older patients. In the context of the potentially new therapeutic landscape, we will also discuss the value and the risk of alternatives to intensive chemotherapy. We believe this is a critical step since availability of non-intensive, effective options raises the issue as to whether intensive chemotherapy remains a choice of significance for older patients with AML.

## 2. Definition of Fitness

The process of assessing older patients’ fitness serves the purpose to determine whether a curative therapy, obtaining a durable complete remission (CR), can be delivered. Such a careful evaluation is intended to exclude therapies that: (1) Can worsen age-dependent frailties, (2) cause organ damage due to pre-existing comorbidities, and (3) the patient is unable to comply with, due to individual characteristics. All these situations can jeopardize short-term life expectancy more than the AML itself, justifying the choice of more conservative (e.g., non-intensive or supportive) options.

Despite the increasing number of scoring systems to help determine the eligibility for intensive chemotherapy (IC), we still lack a universally accepted procedure to define fitness [7]. In this situation of persisting uncertain categorization, patient- (i.e., age and PS) and disease-related factors (i.e., cytogenetics, blood count) are still durably included in these systems to help distinguish patients who can potentially benefit from an intensive approach from those who cannot (Table 1).

A treatment-related mortality calculator based on the outcome of 3365 patients with newly diagnosed AML and submitted to intensive chemotherapy between 1986 and 2009 was published by Walter et al. The calculator, by means of a multiparametric analysis of age, platelet count, percentage of blasts in peripheral blood, albumin level, diagnosis of secondary AML, creatinine, white blood cells (WBC) and PS, was shown to predict the possibility of death within 28 days of induction for patients of all ages [8].

Similarly, Wheatley et al., through the analysis of a series of 2483 patients aged >60 years treated within the UK Medical Research Council trials, developed a prognostic index of survival based on age, PS, cytogenetic risk, and AML type (newly diagnosed vs. secondary), identifying three categories of good, standard, and poor risk with 1 year OS of 53%, 43%, and 16%, respectively [9]. This score, as the one proposed by Walter et al., was originally intended to predict survival and treatment-related mortality in patients submitted to intensive chemotherapy. Consequently, as survival can be considered only a “surrogate” of fitness, the use of these scores in the process of fitness categorization may be questionable.

The National Comprehensive Cancer Network, based on the Comprehensive Geriatric Assessment (CGA) system, has issued specific guidelines for the management of cancer in older adults [10]. Despite predicting survival in hematological patients (through evaluation of cognition, depression, distress, physical function, and comorbidities), CGA represents a complex procedure to carry out; it is time consuming and requires a specialized multi-disciplinary approach involving professionals with specific expertise [11].

Likewise, Deschler et al. analyzed several patient-related variables in the attempt to predict the outcome for older patients affected by AML or MDS. Regardless of disease-related factors, the authors found that the assessment of activity of daily life (ADL) and fatigue were predictive of OS [12].

In order to provide uniform and easily applicable criteria of fitness/unfitness, a panel of experts, convened by the Italian Society of Hematology (SIE), Italian Society of Experimental Hematology (SIES), and Italian Group for Bone Marrow Transplantation (GITMO), selected and summarized a list of 24 conceptual criteria to use in the process of treatment assignment. According to the experts, treatment assignment should recognize three order of therapies: IC, meant as any regimen aimed at achieving CR, for fit patients (F-IC); non-intensive chemotherapy (NIC), meant as any therapy aimed at altering the natural course of the disease but not necessarily achieving CR, for patients unfit to IC (UF-IC); supportive therapy aimed at improving patient’s quality of life without influencing the course of the disease, for patients unfit for non-intensive chemotherapy (UF-NIC) [13]. The operational criteria proposed by the panel were retrospectively applied to a population-based series of 699 patients older than 65 years with a diagnosis of AML, recruited into the clinical trials of the Network “Rete Ematologica Lombarda” (REL). Of 686 evaluable patients, median OS for F-IC, UF-IC, and UF-NIC patients was 10.9, 4.2, and 1.8 months, respectively. Although retrospectively, this analysis showed a significant correlation between fitness and patients’ survival [14]. In our opinion, the SIE, SIES, GITMO score is a simple, reproducible, and easily applicable system to assess fitness and should be considered as a valuable tool to rely on, for the following reasons: (1) As compared to others previously described, this score assesses patients’ conditions and comorbidities in a 360-degree approach, with no exceptions; (2) although requiring a multidisciplinary team to be rated, it doesn’t include frequent evaluations that can alter the patients’ normal life or the physicians’ daily clinical practice; and (3) it can predict outcome regardless of cytogenetic/biological features and specific disease-related characteristics. The score proposed by Ferrara et al. appears particularly suited for the process of treatment selection, aiming at modulating the intensity of the delivered therapy. By contrast, the other available scores try to anticipate the possible outcome that can derive from therapy, once the treatment choice has already been made.

Despite the retrospective application of SIE, SIES, GITMO score by the REL network showing a significative relation between fitness categorization and OS, there is a non-negligible rate of discordance (20.6%) between the “operational criteria” and the actual treatment delivered that may have influenced the results [14]. Therefore, a proper validation of the SIE, SIES, GITMO consensus, requires a prospective trial.

## 3. Biological Features

Recently, the results of comprehensive gene mutation analysis by next-generation sequencing (NGS) have clarified the process of AML development through the demonstration of mutual interactions between genes related to cell proliferation and differentiation, and accumulation of mutations in pluripotent stem cells. The presence of clonal hematopoiesis, defined as the presence of mutations in peripheral blood in the absence of myeloid malignancies, is an age-related process, representing a pre-malignant state that can be triggered by the exposure to cytotoxic damage and rapid hematopoietic stem cell expansion [15]. Intriguingly, in a series of 212 cases of AML and 212 matched controls, clonal mutations could be demonstrated at a median of 9.8 years before the diagnosis of AML. In this study, mutations in *TP53*, *IDH*, spliceosome genes, *TET2*, and *DNMT3A* were significantly associated with AML development [16]. However, routine assessment at diagnosis and during monitoring in patients with AML remains controversial, because of the uncertain clinical significance of these mutations.

Despite the unclear role of molecular findings in the elderly, the emergence of specific gene mutations involved in the leukemic hematopoiesis seems to be age-dependent. For example, the analysis of the frequencies of mutations of *NPM1*, *FLT3-ITD*, *FLT3-TKD* and *CEBPA* in a cohort of 1321 adult patients of all ages with AML, has shown a significant decrease of *NPM1* and *FLT3* mutations with age, with a significant impact on CR rates in older patients (75–86% for patients <60 years vs. 55–63% for patients >60 years) [17]. In addition to that, particular disease-specific biological characteristics of AML associated with poor prognoses, such as p53 gene mutations [18] and complex and/or monosomal karyotype, appear to be more frequent among older patients, justifying the worse outcome even among the fit ones [19,20].

Moreover, the epidemiology of AML is different in older adults if compared to younger populations, with an increased incidence of secondary AML after hematologic and non-hematologic malignancy among older patients [21].

Based on the assumption that any molecular alteration is a potential therapeutic target, an in-depth comprehension of the biology of AML is paving the way to an ever faster drugs development and approval. Therefore, the growing evidences of the biological age-related AML heterogeneity, coupled with the recent approval of the new-generation agents and the availability of accurate, practical, and user-friendly systems for fitness evaluation, could help physicians abandon nihilist solutions that historically resulted in the exclusion of older patients from potentially curative therapies. 

## 4. Treatment Options

### 4.1. “Classic” Intensive Chemotherapy

Since the 1970s, the “7 + 3” regimen (7 days of cytarabine and 3 days of an anthracycline infusion) has remained the paradigm of the “curative-intent” standard of care for patients with newly diagnosed AML [22]. Over the last 40 years, in order to improve the outcome of the “7 + 3 regimen”, several attempts have been made, including increasing the dose of anthracyclines, adding new drugs, or modifying the treatment schedule [23].

In 2009, Lowenberg et al., in a series of 813 patients older than 60 years, reported that the dose of 90 mg/m^2^ of daunorubicin in combination with cytarabine was associated with a statistically significant increase of CR rate from 54% to 64%. However, no significant difference in terms of 2-years OS was observed between 90 mg/m^2^ and 45 mg/m^2^ [19]. Furthermore, the National Cancer Reasearch Institute (NCRI) AML17 trial compared daunorubicin 60 mg/m^2^ versus 90 mg/m^2^, showing no significant differences in term of 2-years OS (60% vs. 59%) and CR rates (75% vs. 73%) [24].

Idarubicin or mitoxantrone have been frequently used in the attempt to substitute for daunorubicin, but, when delivered at the same equitoxic doses, no differences were observed in terms of efficacy [25].

As in younger adults, also in older ones entering CR, delivery of consolidation therapy is regarded as a necessary step to prevent relapses. As of today, despite several trials investigating different consolidation regimes, there is no established consensus or guideline indicating the best consolidation option to offer to older patients with AML. However, for selected “very fit” older adults, it is recommended that high-intermediate dose cytarabine remains the backbone of any consolidation approach. The need for a meticulous selection process for the identification of older patients to treat with high-intermediate dose cytarabine was confirmed in the Cancer and Leukemia Group B CALGB 8525 trial. In that trial, different consolidation regimens were confronted, including 4 courses of cytarabine monotherapy at different doses (standard dose of 100 mg/m^2^ per day continuous infusion for 5 days; intermediate dose of 400 mg/m^2^ per day continuous infusion for 5 days; high dose of 3 g/m^2^ every 12 h on days 1, 3, and 5). For patients older than 60 years, the probability of remaining disease-free at 4 years in each of the three groups was less than 16%. Moreover, treatment-related mortality was particularly high among patients treated with high-dose cytarabine [26].

Notwithstanding, intensive chemotherapy could be theoretically considered the best therapeutic option because of its higher rates of response, the “more is better” strategy is not always acceptable, especially for older patients. Accordingly, the awareness that an intensive approach is likely to lead to life-threatening toxicities rather than to satisfactory results has always represented a factor that discouraged the use of IC in older patients with AML. As a consequence, starting curative-intended therapies without reasonable possibilities to complete the entire course of chemotherapy avoiding unacceptable toxicities is a risk that could be greatly reduced through a precise baseline fitness assessment.

### 4.2. CPX-351, Not the “Same, Old Drugs”

Another way for continuous improvement consists of investigating the efficacy of new formulation of old drugs such as the ones included in the “7 + 3” schedule. CPX-351 represents the paradigm of such efforts. CPX-351 is a new liposomal formulation that encapsulates cytarabine and daunorubicin in a fixed molar ratio of 5:1. CPX-351 showed an “in vivo” superior antileukemic activity than the two free-drugs association [27,28]. The encapsulation of cytarabine and daunorubicin within liposomes increases the magnitude and duration of drug exposure, preserves the 5:1 molar ratio until delivery to the target leukemia cell, and potentially increases the specificity of drug delivery to the leukemia cells [29,30].

A Phase II randomized study comparing CPX-351 versus “7 + 3” in patients with newly diagnosed AML aged 60–75 years showed higher response rates in the CPX-351 group compared to the control arm (66.7% vs. 51.2%, respectively). CPX-351 treatment resulted in more frequent hematological toxicities and a longer median time to count recovery after induction, but in a significantly lower 60-day mortality rate compared to the “7 + 3” regimen (4.7% vs. 14.6%). Incidentally, among those who received CPX-351, a significantly longer OS was observed in a subset of patients with secondary AML (12.1 vs. 6.1 months of control arm, *p* = 0.01) [31]. Based on these findings, 309 older patients (median age 67.8 years) with secondary AML were enrolled in a Phase III study of CPX-351 vs. a conventional “7 + 3” regimen. CPX-351 significantly improved median OS as compared to the “7 + 3” control group (9.56 vs. 5.95 months; hazard ratio, 0.69; 95% Confidence Interval CI, 0.52 to 0.90; one-sided *p* = 0.003). Furthermore, overall remission rate was also significantly higher with CPX-351 than with “7 + 3” (47.7% vs. 33.3%, two-sided *p* = 0.016). Again, the 60-day mortality rate of patients belonging to the CPX-351 arm was significantly lower than in the control arm (13.7% vs. 21.2%). This could be explained by fewer recurrences in the CPX-351 arm, whereas toxicity rate was equivalent. However, the median rate of adverse events per patient-year was 75.68 with CPX-351 versus 87.22 with 7 + 3. Finally, the most frequently reported grade 3 to 5 adverse events in the CPX-351 and 7 + 3 cohorts were febrile neutropenia (68.0% vs. 70.9%), pneumonia (19.6% vs. 14.6%), and hypoxia (13.1% vs. 15.2%) [32].

Based on this finding, CPX-351 was recently approved by the U.S. Food and Drug Administration (FDA) and the European Medicines Agency (EMA) “for the treatment of adults with newly diagnosed, therapy-related acute myeloid leukemia or AML with myelodysplasia-related changes (AML-MRC)” and promises to become the new standard of care for induction therapy in older fit patients with AML.

The remarkable efficacy of this drug combination in the subset AML-MRC may represent a response to the unmet clinical need constituted by the lack of an effective treatment to a specific category of diseases characterized by intrinsic chemo-resistance. Therefore, the safer profile of CPX-351 as compared to standard “7 + 3” offers the possibility to deliver a curative-intended chemotherapy without necessarily intensifying the therapy given.

### 4.3. Allogeneic Hematopoietic Stem Cell Transplantation

Allogeneic hematopoietic stem cell transplantation (HSCT) is clearly the most effective therapy to overcome resistance in AML and to obtain durable remission. Thanks to the modern delivery of reduced-intensity or nonmyeloablative conditioning regimens, the range of age suitable for HSCT has been extended up to 70–75 years [33]. However, only about 8% of older patients with AML are submitted to HSCT each year in the US, due to concerns of high rates of transplant-related mortality and/or relevant impairment of quality of life [34].

Therefore, in pondering the HSCT option, especially for those in whom it represents the only chance for a cure, it is crucial to select patients with a satisfactory fitness status. This assumption brings, again, the need for effective and reliable tools for fitness assessment. The Hematopoietic Cell Transplantation-Specific Comorbidity Index score (HCT-CI), built upon the Charlson Comorbidity Index, is a prospectively validated score, able to predict nonrelapse mortality and OS after HSCT, through a comprehensive analysis of a patient’s physical and mental health [35,36]. The effectiveness of the HCT-CI score is confirmed if one considers that the median OS for HCT-CI of 0 is 45 weeks, while for patients with a score’s level of 3, OS is of 19 weeks, underlying the importance of careful patient selection for transplant [37].

Beyond comorbidities, chronological age, poor PS, and frailty (a syndrome of unintentional weight loss, exhaustion, weakness, decreased walking speed, and decreased physical activity) [38], have been associated with an inferior outcome for older patients submitted to HSCT [39]. Moreover, comprehensive geriatric assessment is currently under investigation in assessing patients’ vulnerabilities before HSCT and could be a useful tool to select those patients more likely to benefit from the transplant procedure. Regarding transplant efficacy, Aoki et al. reported encouraging disease-free and overall survival rates for selected patients older than 65 years, suggesting that age alone had no significant impact on transplant outcome [40]. Similarly, Kurosawa et al., in a registry survey of adults aged 50–70 years, in first complete remission (CR1), showed a longer 3-year OS for those who received HSCT rather than chemotherapy alone (62% vs. 51%, respectively), confirming a survival advantage of the transplant procedure, especially in high-risk patients [41].

Thus far, trials that focus on transplant in older AML patients frequently do not include patients greater than 70 years old. As a result, many conclusions pertaining to transplant that results in AML cannot be definitively applied to patients above that age. Therefore, although HSCT is becoming ever more a reasonably safe procedure, its exact value as a post-remission strategy in older patients is difficult to determine. Furthermore, as HSCT is a complex and aggressive treatment, the inevitable quality of life impairments caused by the transplant procedure need to be weighed against the realistic chances of prolonging survival in a group of patients already disadvantaged by anagraphical factors. For these reasons, the risk of high transplant-related mortality should be taken only when the requirements of “patient’s very high-quality fitness” and “AML high risk of recurrence” are fulfilled.

### 4.4. Hypomethylating Agents

In the last two decades, several new drugs have been investigated to address the need for delivering effective treatment even to those older patients with AML who are considered unfit for standard chemotherapy. Among them, hypomethylating agents (HMAs), with an overall response rate of 25–30%, have become the standard of care for the treatment of older patients with AML (aged ≥60 years) and for those who are considered ineligible for IC [42].

In the AZA-MDS-001 Phase III trial, 113 patients older than 65 years with 20–30% bone marrow blasts infiltration were given azacitidine (AZA 75 mg/m^2^ per day for 7 days) and experienced a 50% 2-year OS versus 16% of those treated with conventional care regimens (IC or low dose cytarabine or supportive therapy). Being randomized in the AZA arm was also associated with fewer total days of hospital staying [43]. In the AZA-AML-001 study, 488 patients with a median age of 75 years were randomized to receive AZA or conventional care regimens (IC or low dose cytarabine). Although no difference in the early-mortality rate was recorded, AZA administration was associated with prolonged median OS compared to conventional care regimens (10.4 vs. 6.5 months, respectively). However, as the AZA-AML-001 only enrolled patients with a WBC count below 15,000/mm^3^, these findings are difficult to extend to include hyperleukocytic AML [44].

In a Phase II multicenter study, 55 patients older than 60 years (median age 74 years) with newly diagnosed AML were treated with decitabine (DEC) 20 mg/m^2^ per day for 5 days. The CR rate was 24%, and it was consistent across the different cytogenetic categories and regardless of the diagnosis of de novo versus secondary AML. The median OS was 7.7 months, with a 30-day mortality rate of only 7% [45]. The following Phase III trial DACO-016 randomized patients between 5 days of DEC every 4 weeks versus low-dose cytarabine or supportive care. The CR rate of patients receiving DEC was 17.8% versus 7.8% of the control group [46]. In the attempt to improve the efficacy of HMAs and to generate remissions of superior quality and longer duration, some clinical trials are evaluating alternative and possibly more optimal dosing schedules of these drugs. By doubling the duration of DEC schedule (20 mg/m^2^ a day for 10 days), Blum et al. reported overall response rates of 64% [47]. An European Organization for Research and Treatment of Cancer (EORTC) Phase III trial comparing a 10-day DEC schedule versus conventional chemotherapy (“7 + 3”) followed by HSCT in patients older than 60 years was recently closed to the enrollment, and the results of the final analysis are eagerly awaited (NCT02172872).

The excellent toxicity profile of all the drugs described above, associated with their proven effectiveness, offers a reasonable therapeutic option to those patients who are considered unsuitable to intensive chemotherapy, without giving up the possibility to obtain a disease-free state. In this subset, fitness assessment becomes crucial to avoid overtreatment for those patients that could benefit from a less intensive approach without renouncing to durable remissions.

### 4.5. Gemtuzumab Ozogamicin, an “Old/New” Target Drug

Monoclonal antibodies are an engineered class of drugs that, by interacting with antigens expressed on the surface of leukemic cells, promote anti-neoplastic activity by immunomodulating tumor microenvironment, exerting cell-mediated cytotoxicity, or by delivering conjugated potent chemotherapeutics to tumor cells [28].

Among different classes of monoclonal antibodies, several new generation antibody–drug conjugates have shown promising results in relapsed/refractory (R/R) AML patients, with the potential to become available in the front line setting as well [48,49,50].

Gemtuzumab ozogamicin (GO), a recombinant humanized anti-CD33 antibody conjugated to a potent cytotoxic agent (calicheamicin), is the only monoclonal antibody approved by the FDA and EMA for the treatment of AML [50]. In May 2010, based on the results of several single-arm Phase I studies, GO was given accelerated approval for the treatment of patients age >60 years with CD33+ AML who were not candidates for aggressive chemotherapy [51]. However, the results of the Phase III post-approval Southwest Oncology Group (SWOG) S0106 trial showed no overall improvement in survival and increased treatment-related mortality in the experimental arm, leading to GO’s withdrawal from the commercial market in October 2010 [52].

More recently, several trials investigated different schedules of GO to reduce toxicity and to maximize efficacy. In the Phase III ALFA-0701 trial, 278 patients with de novo AML (aged 50–70 years) were randomized to receive induction chemotherapy with “7 + 3” (containing daunorubicin at the dose of 60 mg/m^2^ a day for 3 days and cytarabine 200 mg/m^2^ a day for 7 days) plus/minus fractionated GO (3 mg/m^2^ on day 1, 4, and 7 during induction plus one additional shot of 3 mg/m^2^ on day 1 of consolidation course). GO plus IC provided significantly longer median event-free survival (EFS) (19.6 vs. 11.9 months, *p* = 0.00018) and median OS (34 vs. 19.2 months, *p* = 0.046) than “7 + 3” alone. Notably, all patients from the GO arm developed severe neutropenia, thrombocytopenia, and anemia. Moreover, most commonly observed nonhematologic-adverse events were infections (47%) and hemorrhage (18%). As already noted in previous experiences, the clinical benefit was restricted to cytogenetic favorable and intermediate-risk groups [53].

Over the past 6 years, these encouraging results were confirmed in several clinical trials [54,55], leading to refiling and re-approval of GO both by the FDA and EMA for “combination therapy with daunorubicin and cytarabine for the treatment of adult patients with previously untreated, de novo CD33-positive acute myeloid leukemia”.

In the relapsed setting, GO was also effective according to a single-arm Phase II trial, where fractionated GO (3 mg/m^2^ on days 1, 4, and 7) was administered to 57 patients with CD33-positive AML in their first relapse. Consolidation therapy consisted of cytarabine intravenously every 12 h for 3 days. The efficacy of GO was established on the basis of complete remission (CR) rate (26%, 95% CI 16–40%) and duration of remission (median relapse-free remission of 11.6 months) [55]. Accordingly, GO single agent received FDA approval “for the treatment of newly-diagnosed (ND) or R/R CD33-positive AML in adults and in pediatric patients 2 years and older”.

### 4.6. FLT3 Inhibitors

Fms-like tyrosine kinase 3 *(FLT3)*, is a cytokine receptor (CD135) belonging to the receptor tyrosine kinase class III, involved in the proliferation, differentiation, and apoptosis of hematopoietic cells. In AML, two different types of *FLT3* mutations are recurrently found: *FLT3* internal tandem duplications *(FLT3-ITDs)* of the juxtamembrane domain occurring in about 25% of patients, and point mutations in the tyrosine kinase activating loop of the kinase domain *FLT3-TKD* (typically at codons D835 and I836), observed in about 5–10% of patients [56]. FLT3 mutations are detectable in up to 20% of patients older than 65 years. Both mutations cause a ligand-independent activation of the receptor, leading to constitutive activation of the kinase promoting cell growth, survival, and antiapoptotic signaling [57,58]. However, while the *FLT3-ITD* mutations have been associated with adverse prognosis [59], the impact of *FLT3-TKD* mutations remains less clear [60].

To date, several promising *FLT3* inhibitors are being investigated in this particularly unfavorable molecular subtype of AML. *FLT3* inhibitors are classified in first- (sorafenib, midostaurin, leustartinib) and second-generation (quizartinib, crenolanib, and gilteritinib) agents, based on their specificity for *FLT3* [61].

Both aorafenib (a pan-kinase inhibitor approved in several solid malignancies) and lestaurtinib (a staurosporine analog with a broad spectrum of activity against tyrosine kinases) in combination with standard chemotherapy provided promising CR rates without showing a clear benefit in prolonging OS in patients <65 years with diagnosis of de novo *FLT3*-mutated AML [62,63,64,65].

Midostaurin is an oral, first-generation *FLT3* inhibitor able to target multiple pathways involved in tumor development, due to its broad spectrum of inhibition [66]. In a multicenter, double-blind placebo-controlled, Phase III study that enrolled 717 adult patients (18–59 years of age) with newly diagnosed AML carrying *FLT3* mutations, the association of midostaurin/placebo plus a standard induction chemotherapy followed by midostaurin/placebo maintenance showed a clear advantage for the midostaurin containing arm (EFS, 8.2 months versus 3.0 months, *p* = 0.002; disease-free survival (DFS), 26.7 months versus 15.5 months, *p* = 0.01; OS, 74.7 months versus 25.6 months, *p* = 0.009). Of note, few significant differences were observed between the two treatment groups in the rates of adverse events of grade 3, 4, or 5. The most frequent serious adverse reaction (≥10%) in patients in the midostaurin plus chemotherapy arm was febrile neutropenia (16%), which occurred at a similar rate in the placebo arm (16%) [67]. The results of this trial led to the approval of midostaurin by the FDA and EMA, “in combination with standard daunorubicin and cytarabine induction and high-dose cytarabine consolidation chemotherapy, and for patients in complete response followed by Midostaurin single-agent maintenance therapy, for adult patients with newly diagnosed acute myeloid leukemia who are *FLT3* mutation-positive”. However, the full prescribing instructions for midostaurin advise caution in patients >65 years, limiting its use to those who are candidates to “7 + 3” [68]. Finally, midostaurin is under investigation in combination with HMAs in two Phase II trials for the treatment of older patients with AML, in combination with azacitidine (NCT01093573) or decitabine (NCT01846624), respectively.

Quizartinib is a highly selective, second-generation tyrosine kinase inhibitor, designed to inhibit *FLT3* ITD mutation. In a large Phase II trial, quizartinib as a single agent showed auspicious results, achieving a composite complete remission (cCR) rate of 50% in *FLT3-ITD* positive (R/R) patients with AML [69].

Moreover, the results of a randomized Phase III study investigating R/R *FLT-ITD* positive AML patients receiving either quizartinib or salvage chemotherapy, showed a significantly improved median OS (6.2 vs. 4.7 months, *p* = 0.017) and cCR rate (48% vs. 27%, *p* = 0.0001) for quizartinib [70].

Gilteritinib is another potent and selective dual *FLT3* (to a lesser extent to *FLT3-TKD* than *ITD*) and *AXL* (another tyrosine-kinase receptor that promotes proliferation and survival of AML cells) inhibitor [71]. The Phase 3 ADMIRAL trial assessing oral Gilteritinib 120 mg/QD vs. salvage chemotherapy in adult R/R *FLT3*-mutated AML patients led to the FDA approval for gilteritinib “for the treatment of adult patients who have R/R AML with an *FLT3* mutation”. Gilteritinib approval was based on an interim analysis of the ADMIRAL study, demonstrating a cCR of 21% with a duration of 4.6 months [72]. In an extended analysis of the ADMIRAL trial including 371 patients, the median OS in the gilteritinib group was significantly longer than in the chemotherapy group (9.3 months vs. 5.6 months, *p* < 0.001). In an analysis that was adjusted for therapy duration, adverse events of grade 3 or higher and serious adverse events occurred less frequently in the gilteritinib group than in the chemotherapy group. The most common adverse events of grade 3 or higher in the gilteritinib group were febrile neutropenia (45.9%), anemia (40.7%), and thrombocytopenia (22.8%) [73]. An international consortium convening some of the most distinguished European cooperative groups is running a Phase III trial comparing, head-to-head, midostaurin vs. gilteritinib in association with “7 + 3” in patients with de novo, *FLT3*-mutated AML (NCT04027309). However, similarly to midostaurin, quizartinib and gilteritinib tolerability in patients older than 60 is yet to be proven.

The proved efficacy of these agents, both in combination with chemotherapy or as single agents, is expected to counterbalance the negative impact of *FLT3* mutations. Consequently, the approval of these agents is paving the way to new therapeutic approaches, hopefully able to improve the outcome of AML patients carrying *FLT3* mutations [18], regardless of the intensity of treatment given and of fitness status.

### 4.7. BCL-2 Inhibition

The overexpression of pro-survival *BCL-2* family members has been associated with reduced CR rates, earlier relapse, and inferior OS in patients receiving IC for AML [74,75].

Venetoclax, a highly selective oral *BCL-2* inhibitor, has been shown to induce apoptosis both in AML cell lines and primary patient samples in vitro, and in mouse xenograft models in vivo [76].

In a Phase 1/2b trial, enrolling unfit patients with AML, venetoclax 600 mg in combination with low dose cytarabine achieved a cCR rate of 54% [77]. Venetoclax at 400, 800, or 1200 mg daily was also combined with HMAs (AZA or DEC) in a dose-escalation study enrolling older AML patients (>65 years) not eligible for IC. Irrespective of venetoclax dosage, a cCR of 67% was reached, with a median time to response of two cycles. The cCR rates of 60% and 65% were also observed in patients with adverse-risk genetics and those above the age of 75 years. Furthermore, the median duration of response was 11.3 months, with a median OS of 17.5 months. Common grade 3/4 adverse events (>10%) included febrile neutropenia (43%), decreased WBC count (31%), anemia (25%), thrombocytopenia (24%), neutropenia (17%), and pneumonia (13%) [78].

Thanks to these promising results, venetoclax recently received accelerated FDA approval “in combination with HMAs or low-dose cytarabine for the treatment of newly-diagnosed AML in adults who are age 75 years or older, or who have comorbidities that preclude the use of IC”, competing to become the standard of care for first-line treatment in unfit patients with AML.

Finally, ongoing randomized Phase III trials are assessing the potential benefit of venetoclax in association with low-dose chemotherapies (NCT02993523, NCT03069352).

### 4.8. IDH Inhibitors

Isocitrate dehydrogenase 1 and 2 (*IDH1* and *IDH2*) are metabolic enzymes that catalyze the conversion of isocitrate to α-ketoglutarate (αKG) in the cytoplasm and mitochondria, respectively. Hotspot mutations in *IDH1* and *IDH2* are found in about 10–15% of AML patients older than 60 years. [79].

Novel therapeutic approaches have been developed to target leukemic cells carrying *IDH1/2* mutation, in the attempt to revert the mutations-induced myeloid differentiation defects [80,81].

A selective inhibitor of mutant *IDH1* (ivosidenib) was evaluated for the first time as a single agent in a Phase I dose-escalation and dose-expansion study including 258 patients with *IDH1*-mutated hematologic malignancies. Among the 125 R/R AML patients included into the study, Overall Response Rate (ORR) and CR rates were 41%, and 22%, respectively. With a median follow-up of 14.8 months, the median OS was 8.8 months. Ivosidenib was well tolerated with leukocytosis (29.6%), febrile neutropenia (28.5%), and mild prolongation of the QT interval (24.6%) being the most relevant adverse events (in ≥20% of the patients) [82]. On 2 May 2019, the FDA approved ivosidenib for ND AML with a susceptible *IDH1* mutation, in patients who are at least 75 years old or who have comorbidities that preclude the use of intensive induction chemotherapy.

A potent and selective oral inhibitor of mutant *IDH2* (enasidenib) was tested in a Phase I/II trial, in a series of 239 patients affected by myeloid malignancies (mostly R/R AML), carrying *IDH2* mutations. Enasidenib was well tolerated, with hyperbilirubinemia (12%) and differentiation syndrome (6%) being the most significant adverse events. Among 174 patients with R/R AML, ORR was 40% with a CR rate of 19%. The median time to first response was 1.9 months, with a median OS up to 19.7 months among patients achieving a CR. Even the achievement of responses less than CR provided a clear survival benefit [83]. Based on these findings, enasidenib has been recently approved by the FDA as single agent for the treatment of R/R AML with a susceptible *IHD2* mutation.

A preliminary report on the combination of both *IDH*-inhibitors with AZA showed an encouraging ORR of 70% in unfit untreated *IDH1/2*-mutated AML patients [84]. Furthermore, a Phase III study is currently recruiting older subjects (≥60 years) with *IDH2* mutant AML to compare enasidenib efficacy to conventional care regimens (NCT02577406).

Finally, for *IDH1/2* inhibitors, as for the *FLT3* ones, the possibility to deliver specific mutations-driven therapy could help to taper down treatment intensities without giving up the possibility to obtain deep and durable remissions. As a consequence, in these genetic subtypes, the chance to extend the concept of curative-intended therapies to older unfit patients is not to be excluded, even without resorting to intensive chemotherapy.

### 4.9. Hedgehog Inhibition

The hedgehog (HH)/glioma associated oncogene homolog (GLI) signaling pathway, despite being the main regulator of vertebrate embryonic development, is usually silenced in adult tissues [85]. In addition to an important role in normal embryonic development and adult tissue homeostasis, *HH/GLI* aberrant signaling also seems to be pivotal for several cancer hallmarks like genomic instability, proliferative signaling, replicative immortality, tumor invasion metastasis, inflammation, and immune surveillance evasion mechanisms [86].

The evidence of the overexpression of various *HH/GLI* components among chemotherapy-resistant leukemic cell line provided the rationale for combining chemotherapy with *HH/GLI* pathway inhibitors in myeloid malignancies [87].

Glasdegib is a potent and selective oral inhibitor of the receptor smoothened (a positive regulator of the *HH/GLI* pathway) [85,86,87,88]. In an open-label dose-finding Phase I trial enrolling patients with R/R myeloid malignancies (including 28 patients with AML) who were refractory, resistant, or intolerant to previous treatments, the maximally tolerated dose of glasdegib was established at 400 mg, once daily. Among the 28 AML patients, an ORR 57% was observed, without significant nonhematologic adverse events [89]. In the following Phase I/II trial including previously untreated AML and high-risk MDS patients, glasdegib (200 mg once daily), in combination with either low-dose cytarabine, DEC, or IC, showed cCR rates of 8%, 28%, and 54%, respectively [90]. A subsequent randomized Phase 2 multicenter study investigated low-dose cytarabine plus/minus glasdegib (100 mg oral) in a series of 132 patients with AML or high-risk MDS. The CR rates were 17% for the glasdegib arm versus 2% of the standard arm with a median OS of 8.8 and 4.9, respectively. The most common (≥5%) serious adverse reactions in patients receiving glasdegib with low-dose cytarabine were febrile neutropenia (29%), pneumonia (23%), hemorrhage (12%), anemia (7%), and sepsis (7%) [91].

Based on this study, the FDA approved glasdegib “in combination with low-dose cytarabine, for newly diagnosed AML in patients who are 75 years old or older or who have comorbidities that preclude intensive induction chemotherapy”. Finally, glasdegib is currently under investigation in combination with IC or AZA versus IC or AZA alone (NCT03416179).

## 5. Conclusions

After approximately 40 years of stagnation in the management of this malignancy, several new drugs have been recently approved for the treatment of patients affected by AML, potentially improving outcomes even among older patients (Table 2). Since age alone should not exclude older patients with AML from programs of curative-oriented intensive chemotherapy, fitness evaluation takes on great importance and should be mandatory for all patients at diagnosis. In the process of treatment personalization, fitness assessment should be considered a first critical step able to “influence” rather than just “predict” outcome. Consequently, fitness status has to be critically estimated before any other variable, in the attempt to answer the question as to how physicians can put trust in a treatment which a patient is considered unsuitable to. Baseline, appropriate, fitness evaluation could help define a proper “one-fits-one” strategy, without nullifying any further effort to offer the best therapeutic option to each patient. Of note, fitness assessment should be applied dynamically when there is the suspicion that the burden of symptoms and signs are disease-related rather than due to concomitant comorbidities. In this situation, fitness is to be re-evaluated after a vigorous supportive therapy has been instituted.

In the setting of clinical trials, fitness assessment should be considered along with the other quantitative inclusion criteria, in the attempt to distinguish eligible from ineligible patients. In this subset, where fitness assessment mostly relies on arbitrary evaluations rather than on standardized tools, a precise fitness categorization would represent an additional warranty of the quality and subsequent real-life reproducibility of the collected results.

Unfortunately, the picture is complicated by the lack of a universally recognized scoring system for determining patient eligibility to a specific treatment. Consequently, curative-oriented approaches should be wisely selected rather than drastically excluded for elderly patients with AML. Such a wise, rather than draconian, approach should also be preferred in the light of the ever more extended indications of HSCT, which is increasingly used in older patients thanks to the delivery of less intensive conditioning regimens, and better donor selection and toxicities management. Moreover, we wish to emphasize the importance of a proper genetic definition at diagnosis also in older patients: In a time when a plethora of new targeted agents have been approved (i.e., *FLT3* and *IDH1/2* inhibitors) and new potentially curative options for older patients affected by AML are available (Table 3). Nevertheless, as the tolerability and the long-term efficacy of these new drugs have yet to be proven, we are far from defining a hierarchical algorithm for treatment selection. In this view, the “Beat AML Master Trial”, a pioneering collaborative clinical trial testing several novel targeted therapies for older patients with AML, promises to be a remarkable step forward through the full integration of the “genomic screening” and “treatment assignment” [92]. The “Beat AML Master Trial” also serves as an encouragement to include older patients with AML in modern, well-designed clinical trials, which can be a valid alternative to the supportive care option.

## Figures and Tables

**Table 1 cancers-12-00120-t001:** Prognostic algorithms for fitness assessment.

Study	Prognosticators	Aim	Limits
Walter et al.; J Clin Oncol (2011) [8]	Age, platelet count, percentage of blasts in peripheral blood, albumin level, diagnosis of secondary AML, creatinine, WBC count and PS	To predict 28-days treatment-related mortality after induction chemotherapy	Able to predict mortality, not a proper fitness score
Wheatley et al.; Br J Haematology (2009) [9]	Age, PS, cytogenetic risk and AML type (newly diagnosed vs. secondary)	To predict survival after intensive chemotherapy according to patient- and disease-related factors	Able to predict survival, not a proper fitness score
Deschler et al.; Haematologica, (2013) [12]	PS, Health-related quality-of-life scale, Activities of daily living (ADL), Instrumental activities of daily living (IADL), Charlson comorbidity index (CCI), Hematopoietic Cell Transplantation specific Comorbidity Indices (HCT-CI), Get-up and Go Test, Geriatric Depression Scale (GDS), Mini-Mental State Examination (MMSE), EORTC Quality of life questionnaire, percentage of bone marrow blasts, cytogenetics, IPSS in MDS, peripheral blood leukocytes, hemoglobin, serum LDH, serum creatinine, creatinine clearance and serum albumin.	To offer a comprehensive geriatric/quality of life assessment aside from established disease-specific variables	Lack of information regarding the effects of different treatment intensities on outcomes
Ferrara et al.; Leukemia (2013) [13]	Age, PS, comorbidities	To select treatment intensity based on a multi-organ functional evaluation, regardless of disease-related factors	Lack of prospective validation
NCCN clinical practice guideline in oncology [10]	Application of cognition, depression, distress, physical function and comorbidities scales	To predict survival in older hematological patients through cognitive and physical function evaluation	Time-consuming, requires a multidisciplinary team and procedures outside clinical practice

**Table 2 cancers-12-00120-t002:** Newly approved drugs for the treatment of AML and their possible fields of application according to the SIE, SIES, GITMO criteria.

Drug Name	Mechanism of Action	Indications	Applicability
Gilteritinib	*FLT3* inhibition	Treatment of adult patients who have R/R AML with an *FLT3* mutation (FDA)	Single agent in R/R fit/unfit patients
Gemtuzumab Ozogamicin	Anti-CD33 Targeted antibody	Combination therapy with daunorubicin and cytarabine for the treatment of patients aged 15 years and above with previously untreated, de novo CD33-positive acute myeloid leukemia (EMA)	Combination therapy in ND fit adult patients
		Combination therapy with daunorubicin and cytarabine or single agent for the treatment of ND CD33-positive acute myeloid leukemia (AML) in adults and single agent for treatment of relapsed or refractory CD33-positive AML in adults and in pediatric patients 2 years and older (FDA)	Single agent in ND and R/R fit patients
			Combination therapy in ND fit patients
CPX-351	Cytotoxic drug (daunorubicin plus cytarabine liposomal formulation)	Newly diagnosed, therapy-related acute myeloid leukemia or AML with myelodysplasia-related changes (FDA, EMA)	Single agent in ND fit patients
Venetoclax	*Bcl-2* inhibition	In combination with HMAs or low-dose cytarabine for the treatment of ND AML in adults who are age 75 years or older, or who have comorbidities that preclude the use of IC (FDA)	Combination therapy in ND unfit to IC patients
Ivosidenib	*IDH1* inhibition	ND AML with a susceptible *IDH1* mutation, who are at least 75 years old and are considered unsuitable to IC (FDA)	Single agent in ND unfit to IC patients
Enasidenib	*IDH2* inhibition	Adult patients with R/R AML with a susceptible *IDH2* mutation (FDA)	Single agent in fit/unfit to IC R/R patients
Midostaurin	*FLT3* inhibition	In combination with standard daunorubicin and cytarabine induction and high-dose cytarabine consolidation chemotherapy, and for patients in complete response followed by Midostaurin single-agent maintenance therapy, for adult patients with newly diagnosed acute myeloid leukemia who are *FLT3* mutation-positive (FDA, EMA)	Combination therapy in ND fit patients
Glasdegib	*Smoothened* Inhibition (*HH/GLI* positive regulator)	In combination with low-dose cytarabine, for newly-diagnosed AML in patients who are 75 years old or older or who have comorbidities that preclude intensive induction chemotherapy (FDA)	Combination therapy in ND unfit to IC patients

AML: Acute Myeloid Leukemia; SIE: Italian Society of Hematology; SIES: Italian Society of Experimental Hematology, GITMO: Italian Group for Bone and Marrow Transplantation.

**Table 3 cancers-12-00120-t003:** Ongoing clinical trials for older/unfit patients with AML.

Clinical Trial	Phase	Status	Study Drugs	Setting
NCT03416179	PhaseIII	Recruiting	Glasdegib + Azacitidine vs. Glasdegib + placebo	Previously untreated unfit patients with ND AML
NCT02577406	PhaseIII	Recruiting	Enasidenib vs. Azacitidine or Intermediate-dose Cytarabine	Patients 60 years or older with R/R AML after second- or third-line therapy with a susceptible IDH2 mutation.
NCT03173248	PhaseIII	Recruiting	Ivosidenibb vs. Azacitidine or Intermediate-dose Cytarabine	Previously untreated unfit patients with ND AML and a susceptible IDH1 mutation.
NCT02993523	PhaseIII	Active/not recruiting	Venetoclax + Azacitidine vs. Venetoclax + Placebo	Previously untreated unfit patients with ND AML
NCT03069352	PhaseIII	Active/not recruiting	Venetoclax + low dose Cytarabine vs. low dose Cytarabine	Previously untreated unfit patients with ND AML
NCT01093573	PhaseI/II	Active/not recruiting	Azacitidine + Midostaurin	Previously untreated unfit patients with ND AML
NCT02172872	PhaseIII	Active/not recruiting	10-day Decitabine vs. + Standard chemotherapy followed by Allografting	Fit patients 60 years or older with previously untreated ND AML

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
