# Peer review of "Therapeutic Choice in Older Patients with Acute Myeloid Leukemia: A Matter of Fitness"

_cancers, 2020, doi:10.3390/cancers12010120_

Round 1

Reviewer 1 Report

The manuscript by Palmieri and colleagues (“Therapeutic choice in older patients with acute myeloid leukemia: a matter of fitness”) is a review that covers the important issue of how to decide the best, most-effective treatment strategies for acute myeloid leukemia (AML) patients older than 65 years. This manuscript gives a mostly well-balanced overview of the challenges that have to be considered for optimal treatment of this patient population, especially with regard to the increased risk for comorbidities. The authors nicely present various methods to assess patient fitness and how patient fitness may affect treatment outcome in different settings. The focus of the manuscript gets a little lost in the middle of the review, so here are a few suggestions for the authors to improve this manuscript.

The authors write that there is a “significant decrease of NPM1 and FLT3 mutations with age” (lines 105-106), but devote an entire page to clinically used FLT3 inhibitors (lines 274 – 322). This section should either be greatly shortened or partially rewritten to include more data relevant to older AML patients to keep the focus on the main topic – treatment of AML in the aged.

On line 29, the authors cite references from 2000 and 2009 – are there not more recent references on overall survival of elderly AML patients?

The manuscript is generally well-written and interesting to read. Here are a few instances where the authors could improve their work:

All references are numbered twice in the review file – maybe a problem with the editing system?

Line 57, “el at” should be “et al.”

Throughout the manuscript, gene names should be italicized.

Line 93, “cell” should be “cells”.

Lines 118-120 – this sentence is awkward and should be written differently.

Lines 159-161 – this sentence is awkward and should be written differently.

Line 192, “this findings is” should probably be “these findings are”.

Author Response

We would like to thank the reviewers for their comment and advices, which we try to reply on a point-by-point basis.

Authors

1) Overall survival of older AML patients was updated according to the more recent literature (Ref. 3)
2) Minor misspellings were corrected (line 57, 93, 192)
3) Line 118-120 (now 129-131) and 159-161 (now 170-172) were re-phrased as suggested
4) References’ numbering was edited
5) Gene names were italicized throughout the manuscript
6) We intended to dedicate a detailed paragraph to the FLT3 inhibitors (as to the every recently approved drug described throughout the manuscript), including the most relevant studies that led to the approval these drugs. Our aim is to underline how the evolution of the pharmacopeia is expected to change the unfavorable outcome of AML among older patients

Reviewer 2 Report

The review paper by Palmieri et al. provides a succinct, yet comprehensive review of therapeutic options for elderly patients with AML.  There have been a number of new agents and approaches in this area which the authors have done an excellent job of describing and providing expert commentary on.   A pleasure to read.   The article is very well written and can be published with very minor changes.   Most of  these are some format issues that need to be addressed by the authors/editors which are listed below:

1) Table 1 runs over page 2 and 3 and should be moved forward to page 3

2) Line 328 page 8, Line 377, page 9, Lin 383 page 9:  There are single sentence paragraphs which should be incorporated into  full paragraphs

3)  Table 2: Hard to read because the Drug names do not line up with the indications well.  Add spaces to make the table more readable. 

4) Line 410, page 10.  The sentence "We wish to emphasize...." is a run-on sentence that could be broken into two more succinct sentences to make the authors points here.

Author Response

We would like to thank the reviewers for their comment and advices, which we try to reply on a point-by-point basis.

Authors

1) Table 1 was moved from page 2 to 3
2) We incorporated the underlined sentences into full paragraphs, as suggested
3) Table 2 was fixed
4) The sentence at Line 410 (now 417) was re-phrased

Reviewer 3 Report

This draft for a prospective review article deals with treatment options for older patients with acute myeloid leukemia (AML).

This manuscript is authored by acknowledged experts in their field and generally well written. Nevertheless, I have a number of concerns, as outlined below.

1) The requirement of novelty also applies for review articles and the area of therapeutic choice for AML patients is quite frequently reviewed. Therefore, it definitely would be worthwhile to thoroughly explain the specific reasons, why the authors think it important to add another review article to the existing literature. Although this manuscript accurately describes the state of the art, I have doubts that introducing some concepts of personalized medicine is a sufficient reason for another review of the field.

2) Chapter 4 on “treatment options” occupies approximately two thirds of the text and covers an area that was reviewed as  recently as last year by the same authors in this journal (reference 50). Although the agents are reviewed in different order and in different words, there is necessarily a huge and relevant overlap in reported facts.

3) The concept of patient fitness sounds familiar from chronic lymphocytic leukemia. The authors might want to comment on the possibilities and difficulties of adopting this concept to an acute leukemia.

4) The term “emerging approved therapies” in the abstract and elsewhere is confusing, since approval already implies some degree of acceptance and experience.

5) The formatting of Table 1 does not allow a clear attribution of references to prognosticators.

6) With more than one hundred cited references, referencing is excessive.

Author Response

We would like to thank the reviewers for their comment and advices, which we try to reply on a point-by-point basis.

Authors

Even if the analysis of the newest therapeutic choice for older AML patients has already been performed by several groups of experts in the field, this review mainly focuses on how fitness crucially influences the outcome of older patients with AML. There is an extensive paragraph fully dedicated to several scoring systems for fitness assessment, including some considerations regarding the limits of some scores, and the importance of fitness assessment is constantly remarked in each paragraph. Among these scores, we decided to focus on the one proposed by Ferrara et al. (ref. 15) and on its clinical application on a large multicentric series of patients (ref.16). We intended to propose this score as a standard of care due to its reliability and easy applicability, conversely to the ones previously described. To date, this score has been only presented during meetings and, despite being used by different study groups, it lacks proper visibility.

The reason why we decided to perform a detailed analysis of the recently approved drugs is to underline how the evolution of the pharmacopeia is about to change the very unfavorable outcome of AML of older patients.

In response to the redundancy of the expression “emerging approved therapies” we mitigated it throughout the manuscript according to your advice.

Finally, Table 1 was fixed, allowing a clear comprehension of the references-prognosticators correspondence.

Reviewer 4 Report

I'm afraid English language still needs substantial editing, erroneous use of words  (e.g. page 2, line 48, "patient unable to complain with" should read "comply with" and many more examples throughout the text). The text as a whole does not read very smoothly, can be substantially shortened as well as there are many redundancies! Section 2: I am not sure that Walter et al (ref #8) is really an assessment of fitness as attested by the authors, it predicts trreatment related mortality independently of age and does not aim to assess fitness per se but uses disease variables similarly, Wheatly et al was a retrospective analysis of patients in the MRC trials which assesses risk for AML outcome but does not use fitness to assess type of therapy which should be administered to the patient so is misconstrued in teh context of fitness definition Section 3: the discussion of clonal hematopoiesis while of course intriguing does not make sense here and does not add to therapeutic choices in the elderly and could be omitted. The second to last paragraph were obviously added in response to a reviewer comment but doesn't make sense in the context of discussion of biological features Discussion of which elderly AML patients should be treated with 7+3 is completely missing. A review should offer a perspective and not just summarize publications.....i.e. we would not treat a an elderly patient with complex KT with 7+3 any longer. allogeneic spelled wrong (allogenic) throughout. Section on allo transplant is not very informative in the context of this review, no clear advice offered again which patients are unsuitable for intensive therapy and are given HMA instead? Auhtors offer no clear opinion on this. discussion of CPX-351 should follow directy after 7+3 and before HMA for clarity.

Author Response

We would like to thank the reviewers for their comment and advices, which we try to reply on a point-by-point basis.

English was edited to improve the readability of the manuscript. Section 2 dedicated to fitness assessment, and table 1 as a consequence, now include brief discussions about the limits of the cited scores. Section 3 was remodeled for a better understanding of its role in the 360-degrees evaluation of older AML patients Discussion regarding the use of intensive chemotherapy and transplant was implemented with the addition of the authors’ opinions The CPX-351 paragraph is now placed right after paragraph 4.1 (now renamed “classic intensive chemotherapy”).

Reviewer 5 Report

Major points:

1)     The review is structured well with appropriate focus of measurement/assessment of fitness. As such, would be useful to discuss toxicity of all drugs based on the clinical trial data. Some drugs have these listed, but others do not. This would give the reader an idea of how well these drugs could be tolerated in an elderly population.

2)     Would be interesting to propose how we could evaluate fitness assessment scores in the clinical trial setting. How are we going to test whether these clinical scores can adequately predict whether a patient would do well with intensive treatment?A few comments could be made about this in the discussion.

3)     Would be interesting to have a table listing upcoming trial focusing specifically on elderly/unfit patients.

4)   In the monoclonal antibody section, there is no discussion of GO in the relapsed setting, would include since it is approved in the US for this indication. Also, this drug has proven to be beneficial only in the favorable risk subgroup in the newly diagnosed setting.

5)  Recommend modifying the paper to discuss the differences in approvals between the US and Europe.

6) For giteritinib, more recent studied has been published Perl et al. in the NEJM recently. Would update the paper to reflect this data

7) Ivosidenib is approved in the US as a single agent for newly diagnosed patients that can’t tolerate chemo, would update

8) Drug table needs to be modified accordingly based on above

Minor points:

1)      There are sentence structure and grammatical errors throughout the text that require editing.

2)     For table two, would change it so that we have a better notion of options for unfit versus fit. Can divide based on initial first line treatment versus relapsed/refractory.

Author Response

We would like to thank the reviewers for their comment and advices, which we try to reply on a point-by-point basis.

Major points

Each paragraph of chapter 4 and table 2 now include further information about the side-effects of the drugs described above, for a better understanding of the tolerability of these agents. Authors’ point of view regarding the importance of fitness assessment in the clinical trial setting can now be found in the discussion Table 3 was added to include the ongoing clinical trials for older and unfit patients GO paragraph now include information about its use in R/R patients FDA and EMA approval are now specified for each drug Discussion about gilteritinib was updated according to more recent literature Ivosidenib indications were updated Table 2 was updated according to the corrections mentioned above

Minor Points

English was edited to improve the readability of the manuscript. Table 2 was updated with further informations

Round 2

Reviewer 3 Report

The revised version of this prospective review article mainly deals with treatment options for older patients with acute myeloid leukemia (AML).

The changes in the revised version and the author reply address some minor issues I raised in my comments, but not the main concerns that were listed first (comments no. 1-3).

The authors still do not explain the necessity of a new review article on their topic. Since there is substantial overlap with a recent review article by the same senior author in the same journal (ref. 50), one could probably do without the newer text. It is not strictly against the rules to restate the same facts in a new review article, but of little benefit to readers or the field.   

Author Response

We appreciate this second round of reviews, since better understanding of our work by readers represents a priority for us. We intended this work as a 360-degree panoramic on fitness assessment, including both the biologic and physical aspects that could influence treatment choice in older patients with AML. Even if the topic of new drugs has already been addressed by several authors, this is just one aspect of our manuscript that we mean to contextualize in a broader logical reasoning. If we consider fitness assessment a multiparametric evaluation, therefore including also biological parameters (defining the so called “biological fitness” that inevitably influences treatment choice), we believed that a detailed description of new drugs is unavoidable. Consequently, the paragraph dedicated to the new drugs includes all the works that led to the approval of these new agents and this is the reason why there is an overlap with several previously published manuscripts.

The matter of fitness assessment represents a huge unmet clinical need in AML, considering that to date treatments selection for older patients mainly depends on anagraphic considerations rather than on biological age. Unfortunately, the influence of fitness assessment in AML is underestimated in the absence of consensus guidelines and easy applicable scoring systems. Therefore, we decided to dedicate an extended paragraph to the description of the available scoring systems and by positioning it at the beginning of the manuscript, we intended it as the key to reading the whole paper.

Sincerely

The authors

Reviewer 4 Report

Manuscript is much improved by the new flow and clearer sections as well as more explanation of fitness scores and what they bring to the table.

Still could benefit from a more critical view of pros and cons of each therapeutic option to assist readers in decision making. 

Round 3

Reviewer 3 Report

This third version of a prospective review article on assigning treatment options to older patients with acute myeloid leukemia (AML) on the basis of a fitness score, still reports the treatment possibilities on 6 of 9 text pages, with one rather unlinked preceding page about fitness scores. For the third round of reviewing, the authors have provided their ideas, why this review could be different form the many existing reviews on the topic, but these concepts are not obvious from the manuscript itself. Therefore, further work is required to incorporate these ideas into the manuscript and to link the two manuscript parts as outlined below.

1) The authors convincingly explained the rationale of this review article in their reply to reviewer comments, but this information is still lacking in the manuscript itself. For this purpose, an additional paragraph must be inserted at the end of the introduction that details the perspective of patient stratification according to fitness scores with the goal to improve therapeutic outcome. In this context, it would be worthwhile also to refer to the role that patient stratification according to biological features plays for the assignment of the suitable treatment.

2) Chapter 2 on the fitness of older AML patients does not sufficiently point out that, and for which reasons, the authors prefer one of the listed fitness scores, and why its retrospective application to a patient population looked promising. It seems logical that better fitness correlates with better patient survival, but does not indicate that application of that particular score would be of advantage. If references 15 and 16 are central to the review article, it is necessary to thoroughly explain why. It could be easier to highlight these two references, if the reference list was confined to really necessary entries.

3) Table 1 on fitness scores should also contain the reference numbers.

4) Contrary to what the authors say in their reply, the part on treatment options is insufficiently linked with the concept of fitness scores. In fact, fitness is only explicitly mentioned in the chapter on allogenic stem cell transplantation, where it is an obvious requirement.  For better linking the treatment options to patient stratification, fitness and biological features should be combined.  The links must be clearly recognizable in each of the chapters on different treatment options. In this regard, Tables and Figures could be helpful.

5) “Emerging approved drugs” has not been corrected in the heading of Table 2.

Author Response

According to the reviewer's concerns, we performed specific changes to the manuscript, attempting to harmonize the two main themes of our work: fitness assessment and therapeutic options. We intended to help the readers understand our work as it was originally conceived.

Moreover, we also applied the minor changes required (lack of references on table 1 and correction of table 2 heading).

Sincerely

The authors

Round 4

Reviewer 3 Report

Also the fourth version of a prospective review article on assigning treatment options to older patients with acute myeloid leukemia (AML) on the basis of a fitness score in my opinion does not come up to the criteria for publication in a journal with the impact and standing of “Cancers”. Also this third revision insufficiently addressed the raised issues and the text in its present form is far from appropriate for publication as detailed below.

1) Chapter 4 on “treatment options” occupies approximately two thirds of the complete text and covers an area that was reviewed last year with the same senior author in the same journal (reference 50). For the main part of the new manuscript to fulfil the criterion of novelty, which also applies for review articles, it is not sufficient to rephrase the content of a recent previous publication. The authors claimed to intend to link the description of treatment options with concepts of personalized medicine, but an outline of these intentions and explanation of the purpose of the review is still missing in this fourth version, although I specifically suggested this alteration already in each of my previous comments. It is also an accepted rule in scientific writing that an introduction should state the problem at stake followed by the approach to solve it. The introduction thus lacks the counterpart of a corresponding section of the conclusion. Therefore it is absolutely mandatory for the authors to explain in an additional paragraph at the end of the introduction that follows their eloquent presentation of the medical need, how they envision that treatment outcomes can be improved by personalized medicine. For this purpose, in my opinion stratification according to patient fitness must be enhanced by that according to molecular features.

2) Chapter 2 still fails to explain in a convincing manner, why one specific among several listed concepts of fitness scores would be preferable and why its application could lead to improvements of treatment outcome. Especially the newly inserted text at the beginning and end contains unnecessary enumerations, rhetoric repetitions, empty phrases, but little substance. Reference numbers were introduced into Table 1, but the Table remains a mere numeration of prognosticators without naming specific advantages and disadvantages or providing validation. Despite inserted text, the text still contains the statement that reference 15 is promising for the application of a fitness score. Whereas it  seems logical that better fitness correlates with better patient survival, this does not indicate that its application would be of advantage.

3) For the appreciation of certain targeted drugs, it would be important to indicate the absolute frequencies of certain molecular lesions among elderly AML patients, e.g. FLT3 and IDH mutations, for instance in chapter 3 on molecular features.

4) I acknowledge that passages about the specific roles of different drugs for the treatment of elderly AML patients were inserted into chapter 4. However, the chapter and especially Table 2 remain a rather incoherent collection of text on different agents. For better understanding, it is necessary to insert in Table 2 some notion of the mechanisms or targets of the drugs and of the situations, where they can be applied with the best benefit. In particular the Table should indicate the frequencies of eligible molecular or fitness-score groups.

5)Referencing is excessive and has not been reduced.

6) Although this manuscript contains quite well-written passages, it would profit from linguistic revision. For instance, “instances” and “organs impairments” in lines 17 and 33 of the first page should be replaced by the singular.

Author Response

As an answer to the reviewer's comments, we modified the manuscript as follow:

1. we reduced referencing
2. we implemented both tables as suggested
3. we further explained the reason why we proposed the Ferrara score
4. we reduced chapter 4, eliminating drugs that are not yet approved
5. we added absolute frequencies of FLT3 and IDH1/2 mutations among older patients
6. we modified chapter 3 and 5 according to the reviewer’s suggestions
7. we performed minor linguistic revision as suggested

Round 5

Reviewer 3 Report

Also the fifth version of a prospective review article on the role of fitness scores for assigning modern treatment options to older patients with acute myeloid leukemia (AML) in my opinion is not appropriate for publication in a journal with the impact and standing of “Cancers”. This fourth revision was again delivered incredibly quickly, but again was prepared superficially and must be rated as insufficient.

1) As I had pointed out in each of my previous comments, an outline of the review purpose and content must be given at the end of the introduction, because readers will look for this essential information at exactly this place. Although the end of the introduction is quite well written, any review article would be incomplete with some indication at the beginning of how the rest of the review is organized. A few short sentences for this purpose are still sorely missing, although I had already asked four times for this change.

2) The description of fitness scores in chapter 2 and Table 1 was improved, but still is not convincing. Moreover the authors still did not explain, why the retrospective application of the fitness score described in reference 13 was “promising”. A correlation of fitness score with better survival can be expected, but does not indicate that its application would be of advantage.  

3) It is difficult to recognize structure principles in chapter 4 and Table 2. The authors should find some principles that define the order of the agent classes that are reviewed in the chapter and Table, e.g. mechanism of action, tolerability, curative potential. For instance, I find it awkward that XP-351 is dealt with separated from chemotherapies and that the chapter “monoclonal antibodies” contains only one antibody-drug-conjugate.

Author Response

We would like to thank the reviewers for their comment and advices, which we try to reply on a point-by-point basis.

Introduction was implemented as suggested, with a summary of the paper’s structure. The paragraph dedicated to fitness assessment, and table 1 as a consequence, now include brief discussions about the limits of the cited scores. Each paragraph of chapter 4 and table 2 now include further information about the side-effects of the described drugs. Furthermore, the CPX-351 paragraph is now placed right after paragraph 4.1 (now renamed “classic intensive chemotherapy”), and the paragraph about GO was renamed for better comprehension and coherence.